# Dietary Zinc-Loaded Montmorillonite Supplementation Improves Growth Performance, Diarrhea, Intestinal Barrier Function and Regulating Gut Microbiota in Weaned Piglets

**DOI:** 10.3390/ani13233630

**Published:** 2023-11-23

**Authors:** Mingxing Huang, Jiang Yi, Hua Chen, Yuehui Song, Xinyue Hu, Hua Zhou, Nianhua Zhu

**Affiliations:** 1Jiangxi Province Key Laboratory of Animal Nutrition, College of Animal Science and Technology, Jiangxi Agricultural University, Nanchang 330045, China; hmx102828@sina.com (M.H.); yijiang37@sina.com (J.Y.); yhs1013@sina.com (Y.S.); hxyxh93@sina.com (X.H.); 2College of Animal Science and Technology, Hunan Agricultural University, Changsha 410128, China; chenhua111314@sina.com

**Keywords:** Zn-montmorillonite, zinc oxide, growth performance, gut microbiota, intestinal barrier function, weaned piglets

## Abstract

**Simple Summary:**

Information about the effects of zinc-loaded montmorillonite on pigs is scarce. This present study studied the effects of low-dose zinc-loaded montmorillonite on the growth performance, diarrhea, intestinal function, and microbial communities in weaned pigs, which contributed a novel way to reducing the quantities of ZnO in the diet for weaned piglets.

**Abstract:**

This experiment was conducted to investigate whether low-dose zinc-loaded montmorillonite (Zn-MMT) could be used as a potential alternative for high-dose conventional ZnO in preventing diarrhea in weaned piglets. In total, 180 piglets were randomly divided to receive either of the three treatments, with six replicates per treatment and 10 piglets per replicate. The treatments were the control group (CT), the Zn-MMT group (ZM), and the ZnO group (ZO). Compared with the CT group, the ZM and ZO groups exhibited increased ADG at 14–28 days and during the whole period (*p* < 0.05), and a significantly decreased diarrhea rate during the whole period (*p* < 0.01). The activities of T-AOC and SOD were significantly increased (*p* < 0.05), whereas the MDA level decreased (*p* < 0.05) in the serum and colonic mucosa of Zn-MMT- and ZnO-fed piglets. Dietary supplementation with Zn-MMT and ZnO decreased the contents of IFN-γ, TNF-α, IL-1β, IL-6, DAO, and LPS in the serum and colonic mucosa (*p* < 0.01), and increased the IL-10 level (*p* < 0.01). The relative mRNA expressions of TLR-4, claudin 2, Pbd1, and MUC2 were elevated in the colonic mucosa of the Zn-MMT and ZnO groups (*p* < 0.05). 16S rRNA gene sequencing analysis revealed that the abundances of Proteobacteria and Actinobacteria in the ileum and the populations of Ruminnococcus and Faecalibacterium in the cecum were higher in the CT group than in the other two groups. Collectively, dietary addition of Zn from Zn-MMT was comparable to Zn from ZnO for increasing growth performance, alleviating diarrhea, as well as improving mucosal barrier integrity, and regulating the gut microbiota of weaned piglets.

## 1. Introduction

Piglets often suffer from several psychosocial stressors during the postweaning period, such as environmental, nutritional, immunological, and social stresses [1]. These abrupt changes were associated with induced postweaning diarrhea (PWD) and reduced performance. It has been reported that diets with pharmacological levels of Zn (2000–4000 mg/kg of Zn as ZnO) alleviated PWD [2], increased growth performance [3], and reduced the abundances of conditionally pathogenic bacteria (including Escherichia, Enterobacteriaceae, and Campylobacterales) in weaning piglets [4,5,6]. However, feeding with a high level of Zn from ZnO results in a large amount of Zn excreted and causes an environmental problem, which has been restricted in China or banned in the European Union [7,8]. Montmorillonite (MMT) is an aluminosilicate clay mineral consisting of two external tetrahedral silicon layers sandwiching an internal octahedral aluminum layer, with the crystal structure having many 1 nm micropores or channels [9]. It has a large specific surface area and exhibits good adsorption and cation exchange capacities. Loading other compounds (drug molecules) with intercrystalline channels can change the MMT properties and the way of drug molecules [10]. MMT has recently been reported as a controlled release carrier of various drug molecules and nutrients, such as amino acids [11], vitamin B1 [12], and ibuprofen [13], which are embedded into the MMT structure, and then increased drug stability, improved drug dissolution, and achieved the effect of delaying or targeting drug release.

In order to reduce the quantities of Zn excreted into the environment, it is promising to use ZnO at a much lower level and produce similar benefits from pharmacological levels of ZnO. Using the intercalation technology, ZnO molecules could insert into the space structure of MMT and adsorb on its surface [14], which can change the ZnO release location and rate in the intestine. Previous studies found that MMT–ZnO has novel physicochemical properties [12]. Therefore, this experiment was conducted to investigate whether supplementation with 220 mg/kg of Zn from MMT–Zn could alleviate weaned piglet diarrhea and maintain growth performance comparable to pharmacological levels of Zn (2000 mg/kg of Zn from ZnO). Moreover, the effects of MMT–Zn on intestinal immune function and microbial composition of weaned piglets were investigated.

## 2. Materials and Methods

### 2.1. Animal Treatment and Experimental Design

In total, 180 piglets (Duroc × Landrace × Large White; average initial body weight: 6.2 ± 0.5 kg) weaned at 22 ± 1 day of age were randomly allocated to 3 treatment groups of 6 replicates (pens), each with 10 pigs in each pen. The three groups formed on the basis of diet were (1) the control group (CT): basal diet; (2) the Zn-MMT group (ZM): basal diet + 2.0 g/kg Zn-MMT; and (3) the ZnO group (ZO): basal diet + 2.0 g/kg ZnO. Zinc-loaded montmorillonite (Zn-MMT) contains 11% zinc and 0.3% copper and was provided by Changsha AllNew Biological Engineering Co., Ltd. (Changsha, China). ZnO was provided by Jiangxi Baohua Zinc Industry Co., Ltd. (Nanchang, China), and has a Zn content of 76%.

Meal diets were formulated to meet or exceed the nutrient recommendation of NRC (2012) [15]. Table 1 presents the ingredient composition and nutrient level of the basal diet. The Zn-MMT or ZnO was added to experimental diets by substituting of maize. No medicines or antibiotics were used. Feed and water were provided ad libitum throughout the feeding trial. Pigs (with no feed available after an overnight fasting) were individually weighed on days 1, 14 and 28 of the experiment and feed intake per pen was measured daily to calculate average daily feed intake (ADFI), average daily gain (ADG) and feed conversion efficiency (FCR). The number of piglets with diarrhea in each pen was recorded in the morning and evening, and the diarrhea incidence in each pen was calculated. Diarrhea incidence (%) = sum (diarrhea piglet × number of days with diarrhea)**/**(number of piglets in the pen × number of days of trial) × 100%.

### 2.2. Sample Collection and Preparation

At the end of the trial, 18 piglets were selected, with 6 piglets per treatment (1 pig/pen). Blood was collected from the anterior vena cava of these piglets and then executed. The collected blood samples were centrifuged at 3000× *g* for 10 min at 4 °C. The serum was collected in Eppendorf tubes and refrigerated at −80 °C for further analysis. Then, under an aseptic environment, the abdomen was dissected, and the middle sections of the duodenum and proximal colon tissue specimens were collected. The specimens were washed with saline and placed in 10% tissue fixative for subsequent histological analysis. The ileal, colonic, and cecal contents were collected into lyophilized tubes. Another section of the colon was cut with sterile scissors, and the mucosa was scraped with a slide, numbered, snap-frozen in liquid nitrogen, and then stored in a −80 °C refrigerator. The contents were used for subsequent bacterial community analysis, and the mucosa was used for analyzing the mRNA expression level.

### 2.3. Intestinal Morphology

Duodenal and colonic tissue samples were dehydrated, embedded in paraffin, and cut into sections. The sections were then stained with hematoxylin and eosin. Six well-oriented villi and their associated crypt foci were collected from each section of the stained samples. Villi height and crypt depth in the duodenum and colon were measured using Motic Images Advanced 3.2 software (Motic, Xiamen, China). The villi height divided by the crypt depth gave the V/C ratio.

### 2.4. Biochemical Indices and Concentration of Immunoglobulins and Cytokines in Serum

Serum total protein (TP), albumin (ALB), total cholesterol (TC), triglyceride (TG), high-density lipoprotein (HDL), low-density lipoprotein (LDL), glucose (GLU), alkaline phosphatase (ALP), and alanine aminotransferase (ALT) were measured by the commercially kits (Jiancheng Bioengineering Institute, Nanjing, China) with UV–VIS Spectrophotometer (UV1100, MAPADA, Shanghai, China) according to the manufacturer’s instructions. Levels of immunoglobulin A (IgA), immunoglobulin G (IgG), immunoglobulin M (IgM), interleukin-1β (IL-1β), interleukin-6 (IL-6), interleukin-10 (IL-10), interleukin-12 (IL-12), tumor necrosis factor-α (TNF-α), interferon-γ (IFN-γ), diamine oxidase (DAO), and lipolyaccharide (LPS) in the serum and colonic mucosa were determined using enzyme-linked immunosorbent assay kits (Nanjing Jiancheng Bioengineering Institute, Nanjing, China). The activity of superoxide dismutase (SOD), total antioxidant capacity (T-AOC), and the content of malondialdehyde (MDA) in the serum and colonic mucosa were determined using commercial kits (Nanjing Jiancheng Bioengineering Institute, China). 

### 2.5. Ribonucleic Acid Extraction and Quantitative RT-PCR Analysis

Total RNA in the colonic mucosa was extracted using the TranZol Up Plus RNA Kit (TransGen Biotech Co., Ltd., Beijing, China). The purity and concentration of the obtained total RNA samples were measured using Nanodrop ND-1000 (Nanodrop Technologies, Thermo Scientific, Wilmington, DE, USA). TransScript^®^ Uni All-in-One First-Strand cDNA Synthesis SuperMix for qPCR (TransGen Biotech Co., Ltd., China) was used to perform total RNA reverse transcription. Table 2 presents the primer sequences of target genes. Expression levels of glyceraldehyde-3-phosphate dehydrogenase (GAPDH), occludin, claudin-1, claudin-2, Toll-like receptors (TLR4 and TLR9), mucin (MUC1 andMUC2), porcineβ-defensin (Pbd1 and Pbd2), and transforming growth factor beta 1 (TGFb1) in the colonic mucosa were analyzed using the CFX Real-Time PCR Detection System (Bio-Rad, Hercules, CA, USA). The expression levels of target genes relative to GAPDH were calculated using the 2^−∆∆Ct^ method.

### 2.6. Intestinal Microbial DNA Extraction and High-Throughput Sequencing

Total genomic DNA samples were extracted using the commercial OMEGA Soil DNA Kit (M5635-02) (Omega Bio-Tek, Norcross, GA, USA) and then stored at −20 °C for further analysis. The V3–V4 region of bacterial 16S rRNA genes were PCR amplified using the forward primer 338F (5′-ACTCCTACGGGAGGCAGCA-3′) and the reverse primer 806R (5′-GGACTACHVGGGTWTCTAAT-3′). Pair-end 2250-bp sequencing was performed using the Illumina NovaSeq platform with the NovaSeq 6000 SP Reagent Kit (500 cycles) at Shanghai Personal Biotechnology Co., Ltd. (Shanghai, China). Sequence data were mainly analyzed using QIIME2 (2023.8) and the R package (v3.2.0). ASV tables in QIIME2 were used to calculate alpha diversity indices, such as Chao1 richness estimates, Shannon diversity indices, and Simpson indices, and were presented as box plots. The principal coordinate analysis (PCoA) was performed using the PCoA tool in R language. Taxonomy compositions and abundances were visualized using MEGAN and GraPhlAn. The linear discriminant analysis effect size (LEfSe) was performed to detect differentially abundant taxa across groups based on Kruskal–Wallis and Wilcoxon tests. The threshold for the logarithmic linear discriminant analysis (LDA) score was set at 3.5 for the biomarker.

### 2.7. Statistical Analysis

The GLM models of SPSS (SPSS for Windows, version 25.0, Chicago, IL, USA) and Duncan’s multiple comparisons were used to analyze these experimental data. For the data on growth performance, the pen was used as the experimental unit. For other indices, each pig was regarded as the statistical unit. Results are presented as the means and standard error of the mean (SEM). Statistical significance was considered as *p* < 0.05 and was considered a trend at 0.05 ≤ *p* < 0.10.

## 3. Results

### 3.1. Growth Performance

As shown in Table 3, dietary supplementation with Zn-MMT and ZnO significantly reduced the diarrhea rate in the weaned piglets during the whole experiment period (*p* < 0.01). Compared with the CT group, the ZM and ZO groups exhibited significantly increased ADG at 14–28 and 1–28 days as well as significantly increased body weight at 14–28 days (*p* < 0.05). FCR was significantly lower in the ZO group than that in the CT group at days 14 to 28 and the overall period (*p* < 0.05).

### 3.2. Antioxidant Indices, Immunoglobulins, and Cytokines in Serum

As shown in Table 4, the ZM group exhibited significantly increased TC levels (*p* < 0.05) and significantly decreased serum ALT and ALP activities (*p* < 0.01) compared with the CT group. The ZO group exhibited significantly increased GLU content and decreased urea nitrogen concentrations and ALT and ALP activities in the serum (*p* < 0.05). As shown in Table 5, dietary supplementation with Zn-MMT and ZnO increased serum TAOC, SOD activity, and IL-10 levels (*p* < 0.01). By contrast, it significantly decreased TNF-α, IL-1β, and IL-6 levels, diamine oxidase (DAO) activity, and lipopolysaccharide (LPS) content (*p* < 0.01).

### 3.3. Antioxidant Indices and Inflammatory Cytokines in Colonic Mucosa

Dietary supplementation with Zn-MMT and ZnO increased SOD activity (*p* < 0.01) and decreased MDA content (*p* < 0.01) in the colonic mucosa. Moreover, ZnO increased T-AOC in the colonic mucosa (*p* < 0.01). Compared with the CT group, TNF-α, IL-1β, and IL-6 levels were significantly lower and the IL-10 level was significantly higher in the colonic mucosa of piglets in the ZM and ZO groups (*p* < 0.01). The IFN-γ content significantly reduced in the ZO group (*p* < 0.01). Both the ZM and ZO groups exhibited significantly decreased DAO activity and LPS content in the colonic mucosa of piglets compared with the CT group (*p* < 0.01) (Table 6).

### 3.4. Intestinal Morphology

Dietary supplementation with Zn-MMT or ZnO tended to increase crypt depth and VH/CD in the duodenal tissues of weaned piglets (*p* < 0.10) (Table 7, Figure 1).

### 3.5. Relative mRNA Expression of Tight Junction Proteins, Inflammatory Factors, and Toll-Like Receptors in Colonic Mucosa

Compared with the CT group, both the ZM and ZO groups exhibited significantly upregulated relative mRNA expression of MUC2, Pbd1, and TLR4, whereas the relative mRNA expression of MUC1 and claudin1 was downregulated in the colonic mucosa (*p* < 0.01, Figure 2). The relative mRNA expression of occludin in the colonic mucosa was elevated in the ZO group but reduced in the ZM group (*p* < 0.05).

### 3.6. Diversity and Structural Analysis of Bacterial Community in Ileal, Cecal, and Colonic Contents

We investigated the effect of Zn-MMT and ZnO on the intestinal flora structure of weaned piglets through 16S rDNA gene sequence analysis of ileal, colonic, and cecal microorganisms. The results revealed that the Chao1 and Shannon indices of the piglets’ colon, ileum, and cecum were not significantly different among the three treatment groups (Figure 3A–C). The relative abundance of microorganisms in each gut segment at the phylum and genus levels is displayed in Figure 4A,B. At the phylum level (Figure 4A), dominant bacterial communities in the ileum were Firmicutes, Proteobacteria, Actinobacteria, and Chloroflexi. Dietary supplementation with Zn-MMT and ZnO decreased the relative abundance of Firmicutes and increased the relative abundance of Proteobacteria in the piglet ileum compared with the control diet. Firmicutes and Bacteroidetes were the dominant bacteria in the cecum and colon of the piglets. At the genus level, *Lactobacillus* was the dominant bacteria in the ileal, cecal, and colonic contents (Figure 4B). Compared with the CT group, the relative abundance of *Lactobacillus* decreased in the piglet ileum of the ZM and ZO groups and increased in the piglet cecum and colon of the ZM group. To investigate the relationship between the intestinal flora of the three treatment groups, LEfSe and LDA were performed (Figure 5; LDA > 2.5). Compared with the CT group, the ZM group exhibited a significant increase in the relative abundance of 157 bacteria and a decrease in the relative abundance of 16 bacteria in the ileal content of the weaned piglets (Figure 5A). ZnO treatment increased and decreased the relative abundances of 182 and 9 bacteria, respectively, in the ileum (Figure 5B). Both the ZM and ZO groups exhibited a significant increase in the relative abundance of *Proteobacteia*, *Actinobacteria*, *luteimonas*, *Corynebacterium*, *Devosia*, *Dietzia*, and *SMB53* but a decrease in the relative abundance of *g_Epsilonproteobacteria*, *g_Campylobacterales*, *g_Selenomonas*, and *g_YRC22* in the ileum compared with the CT group.

In the cecal content (Figure 5C,D), compared with the control diet, Zn-MMT treatment significantly increased the relative abundance of *g_Ruminnococcus* but decreased that of *p_chloroplast*, *g_Bacteroides*, *g_Campylobacteraceae*, and *g_YRC22*. ZnO treatment significantly increased the relative abundance of 11 types of bacteria (at the genus level were *g_Ruminnococcus*, *g_butyricicoccus*, *g_subdoligranulum*, and *g_luteimonas*) but significantly decreased the relative abundance of 19 types of bacteria. In both the ZM and ZO groups, the relative abundance of *g_Ruminnococcus* was higher, whereas that of *c_Betaproteobacteria*, *o_Tremblayales*, and *g_YRC22* was lower than that in the CT group. In the colonic content (Figure 5E,F), compared with the control diet, Zn-MMT treatment significantly increased the relative abundance of *g_faecalibacterium*, *g_sutterella*, and *f_Alcaligenaceae*, whereas the relative abundance of 14 species of bacteria decreased. The ZnO group significantly increased the abundance of 10 types of bacteria (at the genus level were *g_Ruminnococcus*, *g_eubacterium*, *g_butyricicoccus*, *g_subdoligranulum*, and *g_luteimonas*), whereas decreased the abundance of 27 types of bacteria. The relative abundance of *g_sutterella* and *f_Alcaligenaceae* increased, but that of *p_proteobacteria*, *p_cyanobacteria*, *o_streptophyta*, *C_chloroplast*, *O_Sphingomonadales*, *f_Sphingomonadaceae*, *O_tremblayales*, and *g_Kaistobacter* decreased in both the ZM and ZO group piglets.

## 4. Discussion

Feeding pharmacologically accepted levels of Zn (2000–3000 mg/kg as ZnO) has effectively solved the problem of PWD in piglets, improved piglet performance [16], and modulated the immune response [17,18] and intestinal microbial diversity of piglets [4,5,6]. After entering the digestive tract, MMT can, on the one hand, form a continuous protective film by sliding and extending the crystal layer structure, reduce the number of rumen fiber-decomposing bacteria (*Shigella* spp.), decrease harmful gas stimulation to the rumen, and effectively block the adhesion of pathogenic bacteria, such as *Hepatobacterium* and *Salmonella* spp., to the cells and displacement of indigenous microbes. On the other hand, MMT has strong adsorption, ion exchange, and dispersion properties because of its special 2:1 layered structure that can adsorb metabolic ammonia, viruses, pathogenic bacteria, and various toxins in the intestinal tract and excrete them with feces, thereby reducing damage to the intestinal mucosa, maintaining the health of the digestive tract, and diminishing the effect of diarrhea. Our study results revealed that dietary supplementation with Zn-MMT and ZnO could significantly reduce the incidence of diarrhea and increase the ADG of weaned piglets compared with the control diet. This indicated that Zn-MMT including 220 mg Zn/kg might exhibit an effect similar to that of traditional ZnO (1520 mg Zn/kg) in preventing diarrhea and improving performance in weaning piglets. MMT is widely used in many countries for the treatment of infectious diarrhea in children [19,20] because of its properties of adhesion to digestive mucus and absorption of toxins, bacteria, and rotaviruses [21]. MMT as a feed additive can effectively prevent piglet diarrhea [22,23], and the recommended amount in the feed of piglets was 1–3% [24]. However, this dose may have some adverse effects on piglets (such as nutritional antagonism and reduced feed intake) [23]. The pharmaceutical industry is attempting to increase the antimicrobial properties of MMT by modifying it with copper or zinc ions [25] or to achieve the sustained release effect by loading it with drug molecules in its interlamellar region [10]. Studies have reported that the antibacterial effect of MMT loaded with Cu and Zn was higher than that of MMT loaded with a single metal ion and can significantly prevent diarrhea and improve the growth performance of piglets [26,27]. Song et al. (2013) reported that Cu-loaded MMT can effectively relieve diarrhea in piglets and its effect is equivalent to the anti-diarrhea effect of aureomycin [28]. Hu et al. (2012) reported that Zn-loaded MMT is no less effective in preventing piglet diarrhea than ZnO, and the dose of zinc can be reduced to one-fourth in Zn-MMT, thereby decreasing environmental pollution caused by high-dose zinc in feed [2]. These findings are in line with our findings that a lower dose of Zn-loaded MMT has an effect similar to that of high-dose ZnO in improving growth performance and preventing diarrhea of weaned piglets. The possible reason is that after passing through the stomach and small intestine, the loaded Zn is protected by MMT molecules and plays its anti-diarrhea effect in the hindgut [2]. Most ZnO is dissolved and passes through the stomach at a low pH or is absorbed in the form of Zn^2+^ in the intestine [6], only a small amount of ZnO flows into the hindgut to exert its anti-diarrheal effect [29,30]. 

To reveal the underlying mechanisms, the levels of antioxidant indicators and inflammatory cytokines were measured in the serum and colonic mucosa. Similar to ZnO, Zn-MMT significantly reduced serum ALT and ALP activities, improved blood metabolism, and benefitted piglet health. Zinc affects copper-zinc SOD, glutathione peroxidase, and metallothionein in the antioxidant system in animals, protecting the animal from free radical damage. At the same time, silicate clay increases the SOD activity, reduces the blood MDA level [31], adsorbs or promotes the cellular breakdown of oxygen radicals, and reduces lipid peroxidation in cell membranes to improve the antioxidant capacity of animals. TAOC is a comprehensive indicator of the antioxidant capacity of all organisms. MMT addition to the diet can improve the TAOC and antioxidant capacity of the liver. In this experiment, dietary supplementation with Zn-MMT and ZnO increased the TAOC and SOD enzyme activities in the serum and colonic mucosa of piglets, decreased their MDA content, and improved the serum antioxidant capacity. Cytokines are small peptide molecules secreted by immune cells and certain specific non-immune cells and include interleukins, interferons, tumor necrosis factors, growth factors, and chemokines. They are important for maintaining the stability of the body’s immune system [32]. Insufficient production of anti-inflammatory factors and excessive secretion of pro-inflammatory factors in piglets after weaning leads to intestinal inflammation and damage to intestinal health [33]. IFN-γ can increase transepithelial permeability and alter epithelial tight junctions [34]. In an inflamed colon, IL-1β, IFN-γ, and TNF-α can prevent Na+ and Cl− absorption by the body, thus increasing fecal water content [35]. Thus, the increase in these cytokine levels in the colonic mucosa may cause mucosal damage and dysfunction and lead to diarrhea. IL-10 acts as an anti-inflammatory cytokine that can protect the intestinal barrier function [14]. Weaning piglets produce various stresses, including a reduction in antioxidant levels and upregulation of inflammatory cytokines (TNF-α, IL-1β, and IL-6) in the intestinal tract [36]. In the present study, the contents of pro-inflammatory factors TNF-α, IL-1β, and IL-6 were lower and the content of anti-inflammatory factor IL-10 was higher in the serum and colonic mucosa of the ZM and ZO group piglets than in the CT group piglets. This result is consistent with the results of previous studies [3,37,38]. Moreover, some studies have shown higher levels of IL-10 in the serum and intestinal mucosa of high ZnO-fed pigs [14,39], which is consistent with our data. 

Quantitative RT-PCR analysis on the immune gene and protein showed that the relative mRNA expression levels of MUC2, procine β-defensin 1 (pBD1), and TLR4 were elevated in the colonic mucosa of piglets fed with Zn-MMT and ZnO. MUC2 and pBD1 are innate immune effectors that are also considered antimicrobial peptides [40]. MUC2 is the main component of intestinal mucin and can form a mucosal barrier against invading intestinal bacteria [41,42]. Deficiency or inadequacy of MUC2 in tissues increases the risk of diseases such as colitis and colon cancer in humans [43]. Gonzalez et al. (2004) reported that diosmectite could protect the epithelium from antigens and upregulate the colonic expression of MUC2 [19]. pBD1 exhibits a strong immunoregulatory ability and modifies the inflammatory response [40]. It also enhances the activity of Gram-negative and Gram-positive bacteria, such as *Salmonella typhimurium*, *E. coli*, and *Clostridium* perfringens, which are associated with diarrhea development [44]. TLRs, one group of PRRs in innate immunity, could activate the body’s immune cell response and produce inflammatory cytokines [45]. The present study demonstrated that TLR4 mRNA expression was increased after ZnO and Zn-MMT treatments, whereas TLR4 mRNA expression was downregulated after high-Zn treatment [41]. Improvement in health and prevention of diarrhea of piglets with many additives, such as sodium butyrate [46], acidifiers [6], *Lactobacillus reuteri*, and probiotics [44], were achieved by increasing pBD expression. MUC2 secretion in the intestinal tract was also upregulated by adding ZnO or attapulgite [47]. Therefore, the changes in the colonic antioxidant capacity, mucin profiles, and immunological traits by Zn-MMT and ZnO might support defense mechanisms and help reduce diarrhea in piglets after weaning.

Improving intestinal mucosal permeability is also considered among the different mechanisms of high zinc for preventing diarrhea [48,49]. DAO activity and endotoxin (LPS) content are crucial markers for measuring intestinal membrane permeability [50] and are strongly related to diarrhea in piglets [51]. DAO is an intracellular enzyme of intestinal epithelial cells with high activity and can enter the plasma through the epithelial mucosa [52]. LPS induces an inflammatory response in the intestinal flora and releases inflammatory substances. In the present study, a decrease in DAO activity and LPS content in the serum indicated an increase in the intestinal permeability of piglets fed with Zn-MMT or high ZnO (1500 mg/kg Zn). The intestinal villi in piglets grow rapidly after birth and are approximately 20 μm high after 1 month. After weaning, piglets change their feeding from breast milk to solid feed, which results in villus atrophy and crypt fossa enlargement [53]. Most reports have shown that weaning-induced small intestinal villus atrophy is reversed by dietary zinc supplementation [54]. However, ZnO does not always affect villi morphology [39,55]. In this study, dietary supplementation with Zn-MMT and ZnO only tended to increase intestinal villus height.

Using 16S rRNA gene sequencing, this study revealed no significant differences in Chao1 and Shannon indices in three gut contents among the three groups. In the ileum, with the addition of Zn-MMT, like ZnO, the relative abundance of Firmicutes and Lactobacillus decreased but that of *Proteobacteria* and *Actinobacteria* increased. This finding is consistent with previous findings [4,55,56,57]. Vahjen et al. (2010) found that high dietary addition of ZnO numerically increased proteobacteria and actinobacteria and decreased the relative abundance of Lactobacillus (from 59.3% to 40.7% at 3000 ppm Zn) [58]. This may be because ZnO generally inhibits Gram-positive bacteria and is insensitive to Gram-negative bacteria, whereas *proteobacteria* and *actinomycetes* are mostly Gram-negative. These results contradict those of previous studies that pharmaceutical ZnO increased the proportion of *lactobacilli* and decreased *proteobacteria* and *actinobacteria* in the gut of weaned piglets [5,39]. Therefore, the impact of ZnO on the intestinal microbiota is controversial [8]. Some researchers have indicated that the effect of ZnO in reducing diarrhea may not be related to an increase in lactic acid bacteria and a decrease in pathogenic *E. coli* [55,56,57]. Interestingly, the proportion of Lactobacillus increased in the ZM group, while the ZO group exhibited decreased cecal and colonic microbiota. This suggests that Zn-MMT may be more effective in maintaining the balance of the intestinal micro-ecosystem than ZnO. The LEfSe results showed that dietary supplementation with Zn-MMT and ZnO significantly changed the microbial species in the ileum, but led to a slight change in the microbial species in the cecum and colon. Both Zn-MMT and ZnO increased the relative abundance of potential probiotics *Faecalibacterium*, *Devosia*, *GMB53*, and *Ruminnococcus* and decreased the relative abundance of *Campylobacteraceae*, *Selenomonas*, *YRC22*, *Kaistobacter*, and other potentially harmful bacteria in the cecum and colon. *Faecalibacterium* is a genus of Gram-positive anaerobic bacteria that produces butyric acid and other short-chain fatty acids (SCFAs) and ferments intestinal dietary fiber, which is beneficial for host health [59]. The increased relative abundance of this bacterium improves body weight and feed efficiency in poultry [60,61]. *Ruminnococcus* is an important member of Lachnospiraceae, which, similar to Enterobacter faecalis, ferments indigestible polysaccharides of other bacteria into SCFAs for use by intestinal epithelial cells. Bacterial populations active in the fermentation process belong to some *Firmicutes* families, such as *Lachnospiraceae* or *Ruminococcaceae* [62]. *SMB53* belongs to *Clodiaceae* and can mainly digest digestive mucus and plant-derived sugars (such as glucose) in the intestine [63]. g_*Campylobacteraceae* is among the opportunistic pathogens that cause acute bacterial diarrhea, and the increase in its relative abundance may be the cause of PWD [64].

## 5. Conclusions

In summary, it is concluded that supplementing weaned piglet diets with Zn from Zn-MMT was as efficacious as Zn from ZnO in improving growth performance, reducing diarrhea and promoting intestinal microbiota and barrier function. Meanwhile, the dose of Zn in the Zn-MMT was lower than that in the ZnO. Therefore, Zn-MMT could act as an effective alternative to ZnO in increasing the growth performance and intestinal health of weaned piglets, decreasing the cost of the swine industry, and protecting the environment.

## Figures and Tables

**Figure 1 animals-13-03630-f001:**
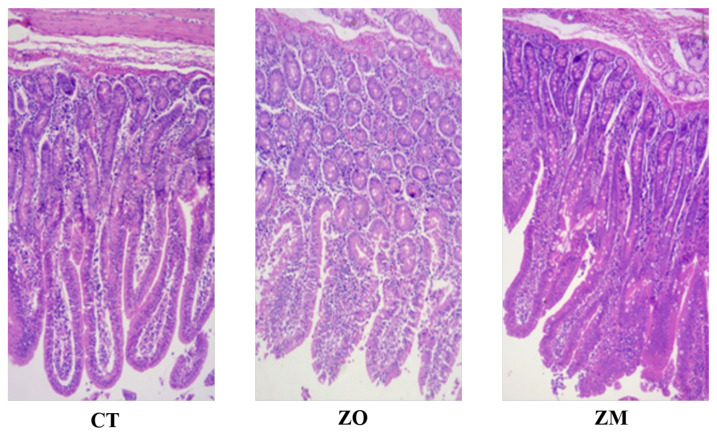
Effects of Zn-MMT on the duodenum morphology of piglets as compared with the control and ZnO. CT, basal diet; ZM, basal diet supplemented with Zn-MMT; ZO, basal diet supplemented with ZnO.

**Figure 2 animals-13-03630-f002:**
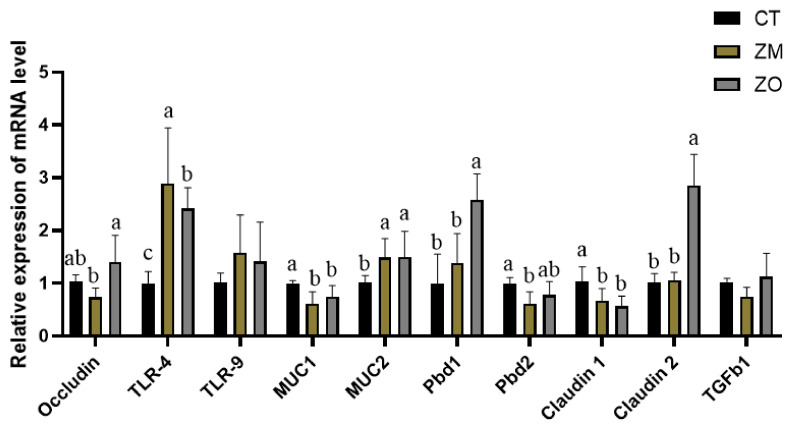
Effects of Zn-MMT on the relative mRNA expression of colonic mucosa barrier functional gene as compared with the control and ZnO. GAPDH was used as an internal standard for normalization. CT, basal diet; ZM, basal diet supplemented with Zn-MMT; ZO, basal diet supplemented with ZnO. abc—means within a row with different superscripts are significantly different at *p* < 0.05.

**Figure 3 animals-13-03630-f003:**
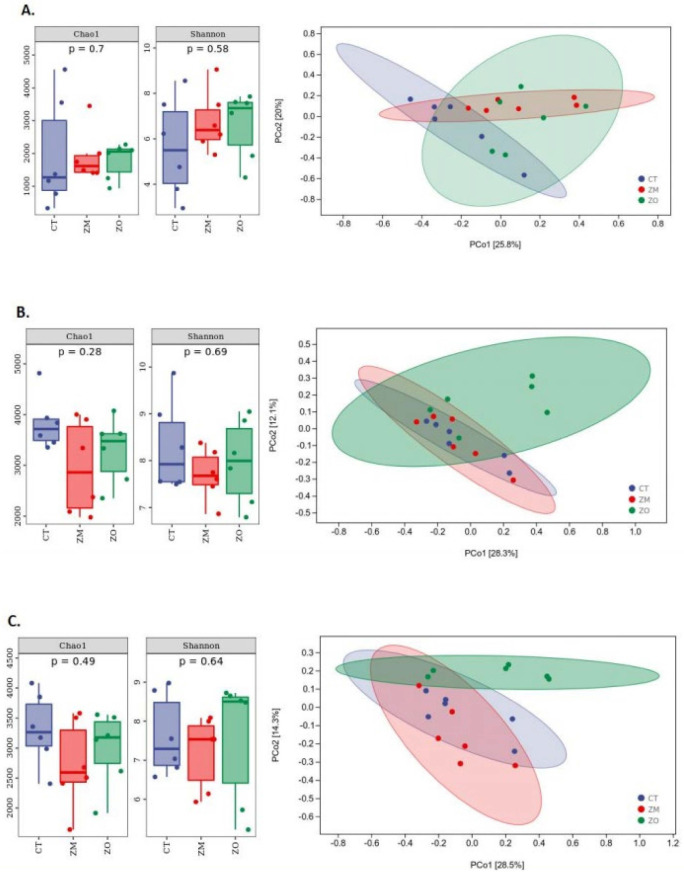
Alpha diversity and PCoA of bacterial communities between the CT, ZM, and ZO treatment groups in the intestine of weaned piglets. (**A**) Colon; (**B**) ileum; (**C**) cecum. CT, basal diet; ZM, basal diet supplemented with Zn-MMT; ZO, basal diet supplemented with ZnO.

**Figure 4 animals-13-03630-f004:**
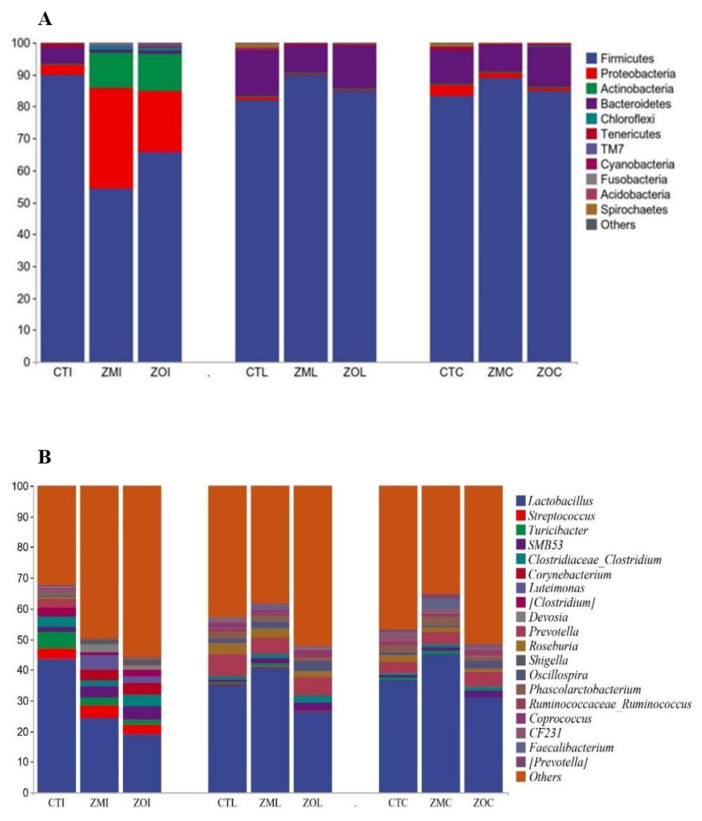
Bacterial diversity of the ileum, colon, and cecum at phylum and genus levels in piglets. (**A**) Relative abundances of the ileal, colonic, and cecal microbiota composition at the phylum level. (**B**) Relative abundances of the ileal, colonic, and cecal microbiota composition at the genus level. CTI, ileum basal diet; CTL, colon basal diet; CTC, cecum basal diet; ZMI, ileum with Zn-MMT; ZML, colon with Zn-MMT; ZMC, caecum with Zn-MMT; ZOI, ileum with ZO; ZOL, ileum with ZO; ZOC, cecum with ZO.

**Figure 5 animals-13-03630-f005:**
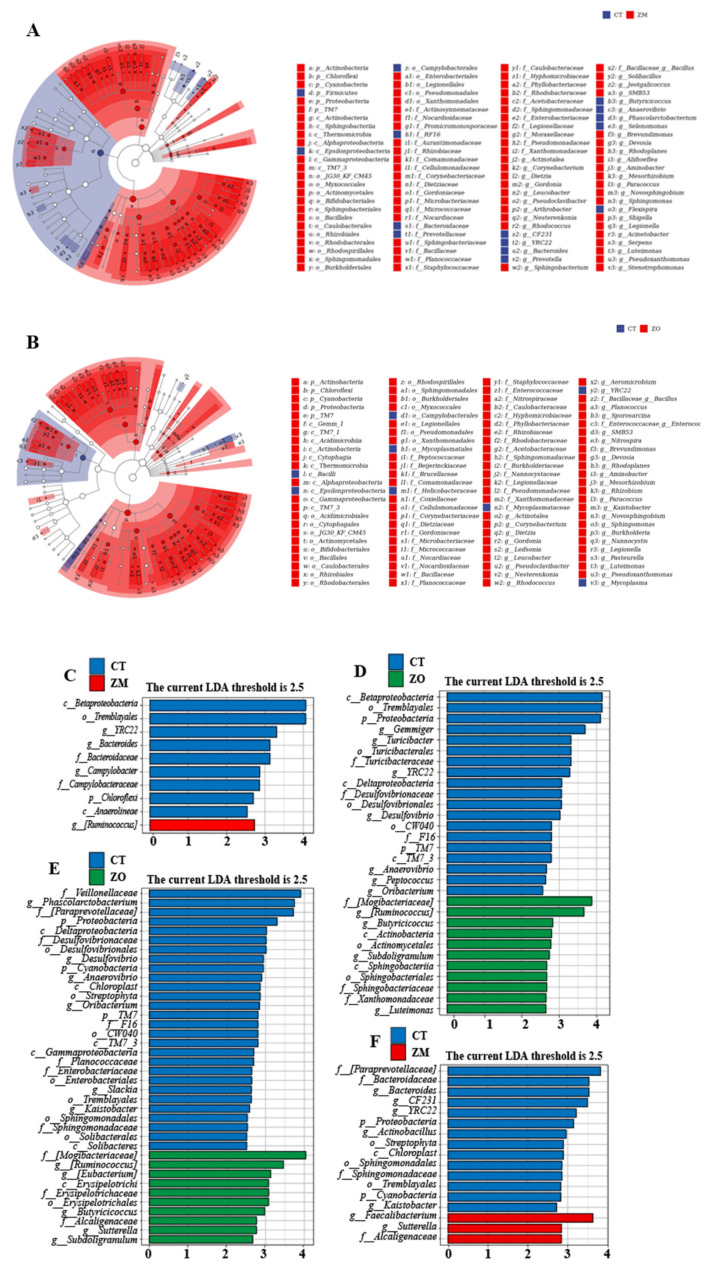
LEfSe analysis and LDA score distribution histogram of effects of Zn-MMT and ZnO on ileal, colonic, and cecal microbial communities of piglets. (**A**,**B**) ileum; (**C**,**D**) cecum; (**E**,**F**) colon. CT, basal diet; ZM, basal diet supplemented with Zn-MMT; ZO, basal diet supplemented with ZnO.

**Table 1 animals-13-03630-t001:** Basic diet composition and nutrient level (%).

Items	Content (%)	Nutrient Levels	
Ingredient		Gross energy (MJ/kg)	14.72
Extruded maize meal	60.00	Crude Protein (%)	20.14
Dehulled soybean meal	22.00	Ca (%)	0.80
Extruded soybean	5.00	Total *p* (%)	0.65
Whey power	2.50	Lysine	1.40
Fish meal	2.50	DL-Methionine + cysteine	0.90
fermented soybean meal	2.00	L-Threonine	0.88
Dicalcium phosphate	0.70	Tryptophan	0.23
Glucose	3.80		
Limestone	0.50		
Premix ^1,2^	2.00		
Total	100.00		

^1^ Premix per kilogram of diet provides: VA, 7500 IU; VD_3_, 750 IU; VE, 25 IU; VK, 2 mg; VB_1_, 1.8 mg; VB_2_, 3.8 mg; VB_6_, 2.2 mg; VB_12_, 0.025 mg; niacinamide, 25 mg; D-pantothenic acid, 16 mg; biotin, 0.18 mg; folic acid, 2 mg; Fe, 100 mg as FeSO_4_; Cu, 25 mg as CuSO_4_·5H_2_O, Zn, 100 mg as ZnSO_4_, Mn, 30 mg as MnSO_4_, I, 0.4 mg as KI, and Se, 0.3 mg as Na_2_SeO_3_. ^2^ According to the zinc content of corn and soybean meal from NRC (2012) [15], it is calculated that the zinc content in the CT group is 120 mg/kg, that in the Zn-MMT group is 340 mg/kg, and that in the ZnO group is 1640 mg/kg.

**Table 2 animals-13-03630-t002:** Primer sequences for real-time quantitative PCR analysis.

Genes	Forword (5′-3′)	Reverse (5′-3′)
GAPDH	TGGAAAGGCCATCACCATCT	ATGGTCGTGAAGACACCAGT
Occludin	CATTATGCACCCAGCAACGA	GCACATCACGATAACGAGCA
Claudin-1	ACACCAAGGCCCTATCCAAA	TTTCTGGTTGTTCCCACACG
Claudin-2	GCATCACCCAGTGTGACATC	GAAGAAGACTCCACCCACCA
TLR4	AAAGTCCAGAATGCGAAGGC	GAACAGAAGTGACCCGGAGA
TLR9	TCCTCTACGACTGCATCACC	CATGCATGTCCAGCTCCTTC
MUC1	ATCTGAGCCCTAGCGAAGTC	CAGGTTCCCACTCCCATCTT
MUC2	CTTGCTCTCGTGTGGAACAG	TCCTCGGGCTTGTTGATCTT
Pbd1	TGCCACAGGTGCCGATCT	CTGTTAGCTGCTTAAGGAATAAAGGC
Pbd2	CCAGAGGTCCGACCACTACA	GGTCCCTTCAATCCTGTTGAA
TGFb1	GCAGGTACTCCTGGTGAACT	AGGATACCAGTCGGGTAGGT

GAPDH: glyceraldehyde-3-phosphate dehydrogenase; TLR: Toll-like receptors; MUC: mucin; Pbd: porcineβ-defensin; TGFb: transforming growth factor beta 1.

**Table 3 animals-13-03630-t003:** Effects of Zn-MMT on growth performance of piglets as compared with the control and ZnO.

Items		CT	ZM	ZO	SEM	*p* Value
Initial body weight (kg)		6.22	6.18	6.17	0.036	0.943
1–14 days after weaning	ADG (g)	278.69	288.1	291.31	1.880	0.096
	ADFI (g)	362.14	365.12	361.55	1.832	0.802
	FCR (%)	1.3	1.27	1.24	0.011	0.302
	Diarrhea incidence (%)	11.19 ^a^	2.74 ^b^	3.45 ^b^	0.821	<0.001
	Weight of 14 days (kg)	10.12	10.21	10.24	0.083	0.666
14–28 days after weaning	ADG (g)	411.91 ^b^	437.86 ^a^	436.43 ^a^	3.863	0.034
	ADFI (g)	618.93	636.31	610.71	4.030	0.101
	FCR (%)	1.50 ^a^	1.45 ^ab^	1.40 ^b^	0.013	0.046
	Diarrhea incidence (%)	5.12 ^a^	1.67 ^b^	1.43 ^b^	0.450	0.001
	Weight of 28 days (kg)	15.89 ^b^	16.34 ^a^	16.35 ^a^	0.069	0.034
1–28 days after weaning	ADG (g)	345.30 ^b^	362.98 ^a^	363.87 ^a^	2.539	0.016
	ADFI (g)	490.54	500.71	486.13	2.487	0.195
	FCR (%)	1.42 ^a^	1.38 ^ab^	1.34 ^b^	0.011	0.019
	Diarrhea incidence (%)	8.16 ^a^	2.20 ^b^	2.44 ^b^	0.590	<0.001

CT: control group; ZM: Zn-MMT group; ZO: ZnO group; ADFI: average daily feed intake; ADG: average daily gain; FCR: feed conversion efficiency; SEM: standard error of mean. *n* = 6. ^a,b^ Means within a row with different superscripts differ (*p* < 0.05).

**Table 4 animals-13-03630-t004:** Effects of Zn-MMT on the blood biochemical indices of piglets as compared with the control and ZnO.

Items	CT	ZM	ZO	SEM	*p* Value
TP (g/L)	49.49	50.75	47.56	0.862	0.333
ALB (g/L)	20.33	20.52	18.27	0.511	0.138
TC (mmol/L)	2.50 ^b^	3.09 ^a^	2.42 ^b^	0.124	0.042
TG (mmol/L)	0.69	0.62	0.52	0.046	0.342
HDL (mmol/L)	0.83	0.90	0.82	0.026	0.425
LDL (mmol/L)	1.41	1.42	1.32	0.050	0.679
GLU (mmol/L)	3.91 ^b^	4.36 ^b^	6.14 ^a^	0.344	0.010
ALT (U/L)	121.96 ^a^	72.11 ^b^	54.94 ^b^	8.745	0.001
ALP (U/L)	294.03 ^a^	200.49 ^b^	172.88 ^b^	16.621	0.002

CT: control group; ZM: Zn-MMT group; ZO: ZnO group; TP: serum total protein; ALB: albumin; TC: total cholesterol; TG: triglyceride; HDL: high-density lipoprotein; LDL: low-density lipoprotein; GLU: glucose; ALP: alkaline phosphatase; ALT: alanine aminotransferase; SEM: standard error of mean. *n* = 6. ^a,b^ Means within a row with different superscripts differ (*p* < 0.05).

**Table 5 animals-13-03630-t005:** Effects of Zn-MMT on blood antioxidant indices, immunoglobulins, and cytokines of piglets as compared with the control and ZnO.

Items	CT	ZM	ZO	SEM	*p* Value
T-AOC (U/mL)	7.53 ^c^	10.31 ^b^	11.51 ^a^	0.422	<0.001
SOD (U/mL)	65.89 ^c^	74.35 ^b^	83.13 ^a^	2.232	0.001
MDA (nmol/mL)	4.48	4.05	3.67	0.148	0.073
IgA (g/L)	2.41	2.50	2.34	0.033	0.128
IgG (g/L)	18.43	18.74	18.11	0.336	0.768
IgM (g/L)	1.44	1.48	1.40	0.020	0.307
IFN-γ (pg/mL)	47.26 ^a^	42.47 ^a^	34.51 ^b^	1.554	<0.001
TNF-α (pg/mL)	70.60 ^a^	54.23 ^b^	48.98 ^b^	2.450	<0.001
IL-1β (pg/mL)	35.52 ^a^	28.91 ^b^	23.50 ^c^	1.275	<0.001
IL-6 (pg/mL)	156.26 ^a^	139.91 ^b^	121.78 ^c^	3.620	<0.001
IL-10 (pg/mL)	12.54 ^c^	14.24 ^b^	15.91 ^a^	0.384	<0.001
DAO (U/mL)	2.64 ^a^	2.40 ^b^	2.13 ^c^	0.054	<0.001
LPS (EU/mL)	0.42 ^a^	0.38 ^b^	0.34 ^b^	0.012	0.003

CT: control group; ZM: Zn-MMT group; ZO: ZnO group; TAOC: total antioxidant capacity; SOD: superoxide dismutase; MDA: malondialdehyde; IgA: immunoglobulin A; IgG: immunoglobulin G; IgM: immunoglobulin M; IFN-γ: interferon-γ; TNF-α: tumor necrosis factor-α; IL-1β: interleukin-1β; IL-6: interleukin-6; IL-10: interleukin-10; DAO: diamine oxidase; LPS: lipolyaccharide; SEM: standard error of mean. *n* = 6. ^a,b,c^ Means within a row with different superscripts differ (*p* < 0.05).

**Table 6 animals-13-03630-t006:** Effects of Zn-MMT on antioxidant and immune-related indices in colonic mucosa of piglets as compared with the control and ZnO.

Items	CT	ZM	ZO	SEM	*p* Value
T-AOC (U/mg.prot)	2.02 ^c^	2.32 ^bc^	2.52 ^a^	0.080	0.026
SOD (U/mg.prot)	10.97 ^c^	13.20 ^b^	15.08 ^a^	0.493	<0.001
MDA (nmol/mg.prot)	1.30 ^a^	1.06 ^b^	0.80 ^c^	0.059	<0.001
IFN-γ (pg/mg.prot)	4.30 ^a^	3.87 ^a^	3.07 ^b^	0.157	0.001
IL-1β (pg/mg.prot)	2.21 ^a^	2.00 ^b^	1.77 ^c^	0.188	<0.001
TNF-α (pg/mg.prot)	7.93 ^a^	6.71 ^b^	6.16 ^c^	0.049	<0.001
IL-6 (pg/mg.prot)	37.53 ^a^	26.83 ^b^	22.18 ^c^	1.630	<0.001
IL-10 (pg/mg.prot)	1.79 ^c^	2.41 ^b^	2.91 ^a^	0.140	0.001
DAO (U/mg.prot)	0.50 ^a^	0.43 ^b^	0.37 ^b^	0.016	0.001
LPS (EU/mg.prot)	0.24 ^a^	0.19 ^b^	0.18 ^b^	0.007	<0.001

CT: control group; ZM: Zn-MMT group; ZO: ZnO group; TAOC: total antioxidant capacity; SOD: superoxide dismutase; MDA: malondialdehyde; IFN-γ: interferon-γ; TNF-α: tumor necrosis factor-α; IL-6: interleukin-6; IL-10: interleukin-10; DAO: diamine oxidase; LPS: lipolyaccharide; SEM: standard error of mean. *n* = 6. ^a,b,c^ Means within a row with different superscripts differ (*p* < 0.05).

**Table 7 animals-13-03630-t007:** Effects of Zn-MMT on the duodenum morphology of piglets as compared with the control and ZnO.

Items	CT	ZM	ZO	SEM	*p* Value
Villus height (μm)	359.19	374.21	412.17	14.72	0.337
Crypt depth (μm)	258.37	287.78	263.29	5.92	0.087
VH/CD	1.38	1.30	1.56	0.048	0.085

CT: control group; ZM: Zn-MMT group; ZO: ZnO group; VH: villus height; CD: crypt depth; VH:CD: villus height: crypt depth; SEM: standard error of mean. *n* = 6.

## Data Availability

The data that support the conclusion of this study will be available from the corresponding author upon reasonable request.

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
