# Peer review of "Dietary Zinc-Loaded Montmorillonite Supplementation Improves Growth Performance, Diarrhea, Intestinal Barrier Function and Regulating Gut Microbiota in Weaned Piglets"

_animals, 2023, doi:10.3390/ani13233630_

Round 1

Reviewer 1 Report

Comments and Suggestions for Authors

Review Report:

The authors investigated the impact of zinc-loaded montmorillonite on weanling pigs. The paper presents a compelling study; however, there appears to be a misunderstanding regarding the study's objectives.

Line 46: Is the primary reason for the European Union's ban on high zinc levels in diets related to environmental excretion? If so, could you clarify the specific indicators you analyzed to address this concern?

Line 82: Could you elaborate on what is meant by "pigs being weighed on an empty stomach"? Did you impose feed restrictions, and if so, how long did these restrictions last?

Table 1: It would be beneficial to explain how zinc was incorporated into the diets in more detail.

Lines 115 to 126: Provide a detailed description of the kits used, including their names and numbers.

Statistical Analysis: Please describe the analytical methods employed for all the results. Did you utilize the same model for all of your data?

Line 169: You mentioned that FCR was significantly lower. Could you specify when this occurred?

Tables: Ensure that the tables are self-explanatory. Describe the treatments used, and include abbreviations that are not present in the journal's list.

Lines 188 and 200: There is inconsistency in your discussion of statistical tendencies. Please provide a clear explanation, addressing when and why tendencies were noted or omitted.

Figure 1: The villi in the Control group appear more intact compared to other groups. What factors might contribute to this observation?

Figure 2: Consider adjusting the color scheme in the graph to enhance differentiation between the groups.

Line 323: Clarify the beneficial role you are referring to.

Results and Discussion: Discuss how these findings can inform industry decisions and their potential implications. Is Montmorillonite (MMT) considered as effective as an antibiotic? Explore the relationship between growth performance, ZnO, MMT, bacterial adversity, and immune-related indices.

Conclusion: Explain the source of information on the dosage of 1520 mg/kg of Zn and why it is significant. This promising data should be further explored to provide more insights.

Author Response

Dear Reviewers:

Thank you for your comments concerning our manuscript entitled “Dietary zinc-loaded montmorillonite supplementation improves growth performance, diarrhea, intestinal barrier function and regulating gut microbiota in weaned piglets” (animals- 2652079). Those comments are valuable and very helpful for revising and improving our paper. We have studied the comments carefully and have made corrections, which we hope meet with approval. Revised portions are marked in red in the paper. The main corrections in the paper and the responses to the reviewer’s comments are as follows:

Reviewer #1:

  1. Comment: “Line 46: Is the primary reason for the European Union's ban on high zinc levels in diets related to environmental excretion? If so, could you clarify the specific indicators you analyzed to address this concern?”.

Response: We are very grateful for your good comment. The European Union's ban on high zinc levels in diets may related to environmental excretion. Feeding with a high dose of zinc will increase the discharge of zinc in manure, we did not analyze the zinc content of pig manure samples, considering that the zinc content of the diets we adopted was 340 mg/kg and 1640 mg/kg respectively (Table1).

  1. Comment: “Line 82: Could you elaborate on what is meant by "pigs being weighed on an empty stomach"? Did you impose feed restrictions, and if so, how long did these restrictions last?”.

Response: We are very grateful for your timely comment. We have revised this sentence. Pigs (with no feed available after overnight fasting.) were individually weighed on d 1, 14, and 28 of the experiment, feed intake per pen was measured daily to calculate Average daily feed intake (ADFI), Average daily gain (ADG) and feed conversion efficiency (FCR).

  1. Comment: “Table 1: It would be beneficial to explain how zinc was incorporated into the diets in more detail”.

Response: Considering your suggestions, we have made an explanation.

The Zn-MMT or ZnO was added to experimental diets by substituting of maize.

  1. Comment: “Lines 115 to 126: Provide a detailed description of the kits used, including their names and numbers.”

Response: According to your comment, we have provided a detailed description of the kits used.

Serum total protein (TP), albumin (ALB), total cholesterol (TC), triglyceride (TG), high-density lipoprotein (HDL), low-density lipoprotein (LDL), glucose (GLU), alkaline phosphatase (ALP), urea nitrogen (UN), and alanine aminotransferase (ALT) were measured by the commercially kits (Jiancheng Bioengineering Institute, Nanjing, China) with UV-VIS Spectrophotometer (UV1100, MAPADA, Shanghai, China) according to the manufacturer's instructions.

  1. Comment: “Statistical Analysis: Please describe the analytical methods employed for all the results. Did you utilize the same model for all of your data?”.

Response: Considering your suggestion, we have revised the Statistical Analysis part.

The GLM models of SPSS (SPSS for Windows, version 25.0, Chicago, IL, USA) and Duncan’s multiple comparisons were used to analyze these experimental data. For data on growth performance, the pen was used as the experimental unit. For other indexes, each pig was regarded as the statistical unit. Results are presented as means and standard error of the mean (SEM). Statistical significance was considered as P < 0.05 and was considered a trend at 0.05 ≤ P < 0.10.

  1. Comment: “Line 169: You mentioned that FCR was significantly lower. Could you specify when this occurred?”.

Response: FCR was significantly lower in the ZO group than that in the CT group during d 14 to 28 and the overall period. We have revised this sentence, and hope to get your approval.

  1. Comment: “Tables: Ensure that the tables are self-explanatory. Describe the treatments used, and include abbreviations that are not present in the journal's list.”

Response: Thank you very much for your detailed advice, we have revised these Tables.

  1. Comment: “Lines 188 and 200: There is inconsistency in your discussion of statistical tendencies. Please provide a clear explanation, addressing when and why tendencies were noted or omitted.”

Response: Thank you very much for your valuable comments, we have revised these two parts.

Dietary supplementation with Zn-MMT and ZnO increased SOD activity (P < 0.01) and decreased MDA content (P < 0.001) in the colonic mucosa.

Dietary supplementation with Zn-MMT or ZnO tended to increase villus height and crypt depth and VH/CD in the duodenal and colonic tissues of weaned piglets (P < 0.10).

  1. Comment: “Figure 1: The villi in the Control group appear more intact compared to other groups. What factors might contribute to this observation?”.

Response: Although the villi in the control group appear more intact compared to other groups, while there was no statistically significant difference. It may be due to individual differences.

  1. Comment: “Figure 2: Consider adjusting the color scheme in the graph to enhance differentiation between the groups.”.

Response: Thank you very much for your detailed advice, this chart is marked with three colors, black, brown and gray, which can distinguish the differences among the three groups. Hope to get your approval.

  1. Comment: “Line 323: Clarify the beneficial role you are referring to”.

Response: We are very grateful for your good comment. We have revised this sentence.

Most ZnO is dissolved and passes through the stomach at a low pH or is absorbed in the form of Zn2+ in the intestine [6], only a small amount of ZnO flows into the hindgut to exert its anti-diarrheal effect [28,29].

  1. Comment: “Results and Discussion: Discuss how these findings can inform industry decisions and their potential implications. Is Montmorillonite (MMT) considered as effective as an antibiotic? Explore the relationship between growth performance, ZnO, MMT, bacterial adversity, and immune-related indices.”.

Response: Thank you for your valuable comments, we have improved the discussion section as much as possible. Notably, montmorillonite (MMT) itself has no antibacterial effect, while loaded zinc has some antibacterial effect, but it is not as effective as antibiotics. The anti-diarrheal effect of ZMM and ZnO may come from modifing intestinal flora, and also improving the convergence and immune function on intestinal mucosa.

  1. Comment: “Conclusion: Explain the source of information on the dosage of 1520 mg/kg of Zn and why it is significant. This promising data should be further explored to provide more insights.

Response: Thank you very much for your comment, we have revised the conclusion part. Hope to get your approval.

In summary, it is concluded that supplementing weaned piglet diets with Zn from Zn-MMT was as efficacious as Zn from ZnO in improving growth performance, reducing diarrhea, promoting intestinal microbiota and barrier function. Meanwhile, the dose of Zn in the Zn-MMT was lower than that in the ZnO. Therefore, Zn-MMT could act as an effective alternative to ZnO in increasing the growth performance and intestinal health of weaned piglets, decreasing the cost of the swine industry, and protecting the environment.

Reviewer 2 Report

Comments and Suggestions for Authors

This experiment was conducted to investigate whether low-dose zinc-loaded montmorillonite (Zn-MMT) could be used as a potential alternative for high-dose conventional ZnO in preventing diarrhea in weaned piglets. The experiments are logical, well organized, and very readable. There are quite a few data available in the manuscript on this topic. Overall, I think the findings are interesting, however, some minor changes are recommended before publication.

·        Specific comments

1. Page 2, What is the basis for adding Zn-MMT? Will the addition of zinc supplements to the treatment group lead to excessive dietary zinc content in weaned piglets?

2. Page 2 line79: Please mention the type of diet. Meal or pelleted?

3. Page 2 line 85: Did you have some preventive strategies against post-weaning diarrhoea?

4. Page 2 line 89 Table 1: Please spell out CP and GE.

5. Check the detection indicators correspond to the data accurately. Such as FCR? UREA?

6. Page 3 line 96: Did you obtain the blood sample before feeding the pigs with experimental diets in adaptation period? It is important to monitor and compare the changes before and after treatments!

7. Line 99, Please provide description of the location from where the intestinal tissue samples were collected.

8. Lines 134-135, Write the full name of the gene.

9. Tables 3-6: Please add the explanation of repetition, group, and P-Value in the footnote.

10. As far as I understood, only one animal per pen was sacrified and sampled for bacterial characterization. It gets not clear in methods, and confirms the reason of using the animal as experimental unit in these parameters.

11. Were caecum and colon sampled separately or mixed during sampling? Authors should describe a bit more in detail which section of colon was sampled (proximal, middle, terminal) why (if it was) was mixed with the caecum to get one simple.

12. Why is there no data on the morphology of the colon in the study?

13. Why were microorganisms detected in the contents of multiple intestinal segments instead of a specific segment? Why were there no multiple group comparisons between LEfSe analysis and LDA score distribution histograms?

14. Line 355, “Quantitative RT-PCR analysis on the immune gene and protein showed……” If there is protein data, please provide additional information.

15. This is a significant issue as there have been multiple occurrences of different ZnO additions in the manuscript. Please carefully review and make appropriate revisions.

Comments on the Quality of English Language

The English writing should be improved due to some spelling and grammar mistakes in this article. 

Author Response

Dear Reviewers:

Thank you for your comments concerning our manuscript entitled “Dietary zinc-loaded montmorillonite supplementation improves growth performance, diarrhea, intestinal barrier function and regulating gut microbiota in weaned piglets” (animals- 2652079). Those comments are valuable and very helpful for revising and improving our paper. We have studied the comments carefully and have made corrections, which we hope meet with approval. Revised portions are marked in red in the paper. The main corrections in the paper and the responses to the reviewer’s comments are as follows:

  • Reviewer #2:
  1. Comment: “Page 2, What is the basis for adding Zn-MMT? Will the addition of zinc supplements to the treatment group lead to excessive dietary zinc content in weaned piglets?”.

Response: We are very grateful for your good comment. The basis for adding Zn-MMT is to improve the utilization efficiency of zinc and reduce the amount of zinc by loading montmorillonite. The treatment of this experiment does not lead to excessive zinc in piglets' diet.

  1. Comment: “Page 2 line79: Please mention the type of diet. Meal or pelleted?

Response: Considering your suggestion, we have revised the description of the diet.

Meal diets were formulated to meet or exceed the nutrient recommendation of NRC (2012).

  1. Comment: “Page 2 line 85: Did you have some preventive strategies against post-weaning diarrhoea?”.

Response: Thank you for your comment. No medicines, antibiotics or other preventive strategies were used in the present study.

No medicines or antibiotics were used.

  1. Comment: “Page 2 line 89 Table 1: Please spell out CP and GE.”

Response: According to your comment, we have spelled out CP and GE in Table 1. Table 1

  1. Comment: “Check the detection indicators correspond to the data accurately. Such as FCR? UREA?”.

Response: Thanks for your thoughtful comments, we have thoroughly reviewed the indicators and data. Table 1 and Table 4

Feed intake per pen was measured daily to calculate Average daily feed intake (ADFI), Average daily gain (ADG), and feed conversion efficiency (FCR).

  1. Comment: “Page 3 line 96: Did you obtain the blood sample before feeding the pigs with experimental diets in adaptation period? It is important to monitor and compare the changes before and after treatments!”.

Response: To be honest, we didn't collect the blood sample before feeding the pigs with experimental diets, because it was during the Chinese New Year holiday and we were short of staff.

  1. Comment: “Line 99, Please provide description of the location from where the intestinal tissue samples were collected”

Response: The middle sections of the duodenum and proximal colon tissue specimens were collected.

  1. Comment: “Lines 134-135, Write the full name of the gene”.

Response: Considering your suggestion, we've rewritten the gene names. Table 2.

  1. Comment: “Tables 3-6: Please add the explanation of repetition, group, and P-Value in the footnote.”.

Response: We are very grateful for your good comment. We have revised these Tables

  1. Comment: “As far as I understood, only one animal per pen was sacrificed and sampled for bacterial characterization. It gets not clear in methods, and confirms the reason of using the animal as an experimental unit in these parameters.”.

Response: Thank you for your sincere suggestion. In fact, we could collect stool samples from each piglet per pen, and mix them into one sample. However, a previous study reported that stool samples could not fully represent the bacterial community in the gut (Zmora et al., 2018, Cell 174, 1388–1405, https://doi.org/10.1016/j.cell.2018.08.041). Therefore, the pig with the average BW of each pen was selected to obtain digesta samples.

  1. Comment: “Were caecum and colon sampled separately or mixed during sampling? Authors should describe a bit more in detail which section of colon was sampled (proximal, middle, terminal) why (if it was) was mixed with the caecum to get one simple”.

Response: We are very grateful for your good comment. In the present study, the cecal and colonic digesta were sampled separately during sampling. Meanwhile, the digesta sample of the proximal colon was collected to determine the community of gut microbiota.

  1. Comment: “Why is there no data on the morphology of the colon in the study?”.

Response: Thank you for your valuable advice, the intestinal morphology of the colon was not determined and we have revised the statement.

Dietary supplementation with Zn-MMT or ZnO tended to increase villus height and crypt depth and VH/CD in the duodenal and colonic tissues of weaned piglets (P < 0.10) (Table 7).

  1. Comment: “Why were microorganisms detected in the contents of multiple intestinal segments instead of a specific segment? Why were there no multiple group comparisons between LEfSe analysis and LDA score distribution histograms?

Response: Our original intention was to find out which segment of the intestine is most affected by Zn-MMT or ZnO. The multiple group comparisons between LEfSe analysis and LDA score distribution histograms have been presented in the text of the results section.

  1. Comment: “Line 355, “Quantitative RT-PCR analysis on the immune gene and protein showed……” If there is protein data, please provide additional information

Response: Unfortunately, we have not measured the protein expression data on the immune gene.

  1. Comment: “This is a significant issue as there have been multiple occurrences of different ZnO additions in the manuscript. Please carefully review and make appropriate revisions

Response: We are very grateful for your comment. We have carefully reviewed and made appropriate revisions to the manuscript.
